# Tracking W-Formate Dehydrogenase Structural Changes During Catalysis and Enzyme Reoxidation

**DOI:** 10.3390/ijms24010476

**Published:** 2022-12-28

**Authors:** Guilherme Vilela-Alves, Rita Rebelo Manuel, Ana Rita Oliveira, Inês Cardoso Pereira, Maria João Romão, Cristiano Mota

**Affiliations:** 1Associate Laboratory i4HB—Institute for Health and Bioeconomy, NOVA School of Science and Technology, Universidade NOVA de Lisboa, 2829-516 Caparica, Portugal; 2UCIBIO, Applied Molecular Biosciences Unit, Department of Chemistry, NOVA School of Science and Technology, Universidade NOVA de Lisboa, 2829-516 Caparica, Portugal; 3Instituto de Tecnologia Química e Biológica, Universidade Nova de Lisboa, Av. da República, 2780-157 Oeiras, Portugal

**Keywords:** formate dehydrogenase, CO_2_ reduction, X-ray crystallography, molybdopterin, tungsten cofactor, redox enzymes

## Abstract

Metal-dependent formate dehydrogenases (Fdh) catalyze the reversible conversion of CO_2_ to formate, with unrivalled efficiency and selectivity. However, the key catalytic aspects of these enzymes remain unknown, preventing us from fully benefiting from their capabilities in terms of biotechnological applications. Here, we report a time-resolved characterization by X-ray crystallography of the *Desulfovibrio vulgaris* Hildenborough SeCys/W-Fdh during formate oxidation. The results allowed us to model five different intermediate structures and to chronologically map the changes occurring during enzyme reduction. Formate molecules were assigned for the first time to populate the catalytic pocket of a Fdh. Finally, the redox reversibility of *Dv*FdhAB in crystals was confirmed by reduction and reoxidation structural studies.

## 1. Introduction

Climate change is a reality hard to ignore in current times, and it will be even more so in the coming future. To that reality, the major contributor is none other than CO_2_, a greenhouse gas of which anthropogenic emissions have been rapidly increasing since the Industrial Revolution with slim prospects of reduction in the future [1]. Therefore, the paramount importance of developing technologies capable of capturing CO_2_ and mitigating this problem is evident. *Desulfovibrio vulgaris* Hildenborough (*Dv*) Formate Dehydrogenase 1 (*Dv*FdhAB) is a W-containing enzyme capable of interconverting CO_2_ and formate under mild conditions, with high specificity and turnover rates (being especially active towards CO_2_ reduction, when compared with other Fdhs) [2]. Thus, *Dv*FdhAB prompts research to better understand its properties and mechanism, aiming at harnessing its catalytic activity towards biotechnological and chemical applications, using either engineered optimized enzymes or bio-inspired, catalytically active, model compounds.

*Dv*FdhAB contains a W active site coordinated by two Molybdopterin Guanine Dinucleotides (MGD), one selenocysteine (SeCys) and one terminal sulfido ligand (=S/−SH). Additionally, four [4Fe-4S] clusters are used for electron transfer. Recent studies [2,3] structurally characterized the oxidized (PDB ID: 6SDR), formate-reduced (PDB ID: 6SDV) and dithionite-reduced (PDB ID: 7Z5O) forms of *Dv*FdhAB, showing important differences between the two redox states and providing some clues about the catalytic mechanism. However, some doubts persist regarding the exact mechanism used by metal dependent Fdhs. It is still under debate whether the W coordination sphere is stable during catalysis and oxidation of formate occurs in the second coordination sphere through a hydride transfer, or if the SeCys ligand dissociates from the metal, allowing formate to bind directly to W [3,4,5,6,7,8]. In addition, a second issue arises, pondering if the hydride transfer occurs to the metal or to the sulfido ligand [3,4,5,6,7,8,9].

The reduction of *Dv*FdhAB can be achieved by soaking or co-crystallizing crystals of the as-isolated protein with sodium formate or sodium dithionite both yielding identical structures, as previously shown [2,3]. Despite the end states being already known and the structural changes between both forms having been characterized, the exact mechanism of how those changes occur remains unclear. In this study, we soaked as-isolated crystals of *Dv*FdhAB with sodium formate and flash cooled them in liquid nitrogen at different time points. Crystal structures revealed that the structural changes following enzyme reduction are sequential, making it possible to track different events in time. Moreover, we confirmed that the enzyme is reversibly operating in the crystals through an oxygen reoxidation step that returns it to the initial oxidized form.

## 2. Results

### 2.1. Time-Resolved Reduction of As-Isolated DvFdhAB Crystals

We performed kinetic assays with *Dv*FdhAB in solution under standard [2] and crystallization conditions (see Materials and Methods section). Previous studies have shown that the enzyme needs to be pre-activated (DTT or TCEP) to have maximum activity for formate oxidation [2]. As shown in Table 1, the activity drops to 0.8% without enzyme pre-activation and in crystallization conditions. The slow kinetics prompted us to design a time-resolved experiment aiming at trapping intermediates of the reaction.

Aerobically isolated and oxidized *Dv*FdhAB was crystallized in an anaerobic environment and crystals were soaked with sodium formate and flash frozen with different time delays. Our crystallization experiments resulted in five crystal structures obtained after crystal soaking periods of 1 min (Fdh_1min, PBD ID: 8BQG), 1 min 30 s (Fdh_1.5min, PBD ID: 8BQH), 3 min (Fdh_3min, PBD ID: 8BQI), 5 min (Fdh_5min, PBD ID: 8BQJ) and 22 min (Fdh_22min, PBD ID: 8BQK). All crystals belong to the same space group (P2_1_2_1_2_1_), and the structures were solved to resolutions ranging from 1.59 Å to 2.36 Å, with good refinement and geometry statistics (Appendix A). The good quality of these models enables a direct comparison with the previously reported as-isolated (oxidized) and formate-reduced forms (PDB IDs: 6SDV and 6SDR, respectively) [2].

This comparative analysis reveals that the experiments covered the full extent of the reduction as the Fdh_22min structure (PDB ID: 8BQK) is essentially identical to the reduced form (PDB ID: 6SDV) with an RMSD of 0.146 Å for 1021 α-carbons. Superposition of the new five structures with the two reference structures (PDB ID: 6SDR and 6SDV) allowed us to track the active site changes and the direction of the alterations (towards the oxidized or the reduced forms).

Comparing the first two structures, Fdh_1min and Fdh_1.5min, with the as-isolated form (PDB ID: 6SDR), we found small differences between the refined models. The active site conformation of Fdh_1min and Fdh_1.5min is similar to the oxidized form. Notably, in Fdh_1min, one formate molecule is observed for the first time close to the active site, making hydrogen bonds with Nε from R441 and Oγ from T450 (Figure 1a).

Fdh_1.5min crystals diffracted to higher resolution, but a formate molecule could not be modelled in the position found in Fdh_1min, although, a modelled water molecule does not completely explain the density, meaning that it can be a mixture of formate and water molecules. However, at the entrance of Fdh_1.5min pocket it was possible to model two formate molecules (Figure 2a) hydrogen bonded to S194 and H457 (not shown). One of the features in the reduction of *Dv*FdhAB is the conformational change in MGD2 with the distortion of the ribose moiety [2]. Here, we observed this conformational change in the first two time-points structures (Figure 1b and Figure 2b), whereas the remaining active site residues are in the oxidized form conformation, suggesting that the conformational change in MGD2 is the first event in the reduction process.

Fdh_3min shows a major structural change with the shift of the loop I191-P198 and H193 sidechain in a new conformation (Figure 3). H193 changes from the oxidized to the reduced conformation between 1.5 min (Fdh_1.5min) and 5 min (Fdh_5min), exhibiting a double conformation at 3 min (Fdh_3min). After 5 min the histidine conformation is stable in the reduced form position (RMSD of 0.11 Å for 10 atoms) (Figure 4a). The adjacent loop I191-P198 (which includes the W coordinating SeCys192) is modelled in an intermediate position between the oxidized and reduced forms in Fdh_3min (Figure 3a), suggesting that this loop moves together with H193. At 5 min (Fdh_5min) this loop superimposes the reduced form, along with the W ligands (Sulfido group, SeCys192 and MGD2 dithiolene moiety) (Figure 4).

On the other hand, the MGD2 ribose moiety in Fdh_3min and Fdh_5min is modelled in a conformation between oxidized and reduced forms (Figure 3b and Figure 4b).

The last conformational change happening in the reduction of *Dv*FdhAB is the concerted shift of E443, Q890 and MGD2. For these residues (E443 and Q890), despite small shifts, the structures until 1.5 min are comparable to the oxidized form. Then, from 3 min onwards the conformation of E443 and Q890 gradually shift to the positions found in the reduced form (Figure 3b). In Fdh_5min we can see a major shift in the position of E443 (1.44 Å (Cδ-Cδ) when compared with the as-isolated form), but the structural change does not correspond to the rotamer that we observe in the reduced state, it is a tilt of the side chain (Figure 4b and Appendix A). This conformation is unusual but probably promoted by a slight rearrangement of the active site or protonation of E443. Nevertheless, it seems that this rearrangement is happening before Q890 changes to its reduced form conformation.

Fdh_22min shows that at this point, the protein reaches the reduced form conformation, and the Fdh_22min active site closely matches the reduced reference (Figure 5a,b). The MGD2 is now superimposable with the reduced form (RMSD of 0.12 Å for 47 atoms) (Figure 5b). At this stage, we assumed that the enzyme is fully reduced and blocked in this state due to the lack of electron acceptors.

### 2.2. Reoxidation of Formate Reduced DvFdhAB

*Dv*FdhAB is expected to reversibly alternate between the oxidized and the reduced states as that is a fundamental requirement for its catalytic activity. Nonetheless, the structure of a reoxidized *Dv*FdhAB has never been characterized. We now report the structure of *Dv*FdhAB co-crystalized with sodium formate and reoxidized by exposure to atmospheric oxygen for 12 min (PDB ID: 8BQL). The structure obtained belongs to the space group P2_1_2_1_2_1_ and was solved at 1.91 Å resolution, to a crystallography R_work_ and R_free_ of 18.8% and 22.5%, respectively, while presenting good geometry statistics (Appendix A).

Analysis of this structure shows that after 12 min exposure to oxygen the protein active site is fully oxidized, being identical to the previously reported as-isolated form (RMSD of 0.159 Å for 1062 α-carbons). All the amino acids that had changed their positions return to the oxidized form conformation (e.g., RMSD of, respectively, 0.20 Å for H193 (for 10 atoms), 0.17 Å for U192 (for 6 atoms) and 0.25 Å for the Active Site: W, MGDs and Sulfur co-factor (for 94 atoms) (e.g., RMSD of, respectively, 0.20 Å for H193 (for 10 atoms), 0.17 Å for U192 (for 6 atoms) and 0.25 Å for the Active Site: W, MGDs and Sulfur co-factor (for 94 atoms). Finally, no signs of oxygen-related damage were visible either in [4Fe-4S] centers or in the W active site (Figure 6).

We also tried to perform a time-resolved experiment in the reoxidation direction; however, due to experimental limitations, such as controlling O_2_ diffusion, the results were not conclusive. We obtained data that showed a mixture of states without a clear track of the sequential events.

## 3. Discussion

When *Dv*FdhAB is purified under aerobic conditions it is in an oxidized state that needs to be pre-activated to yield maximum activity [2]. In our experiments, we used the enzyme without any pre-activation, allowing us to obtain slower kinetics (0.8% of maximum activity, Table 1) and to monitor the structural changes over time. Our results show the time-resolved reduction of *Dv*FdhAB with formate, for which only the starting and end points were known [2], allowing us to follow the chronological order of the structural changes observed between the two states. Despite the relatively low time resolution (minutes), as the procedure was performed by hand, we were able to establish a general order of events, enabled by the slow kinetics under these conditions. Even though the use of macrocrystals may cause diffusion effects, which impact the synchronization of the reaction starting point, the fact that the crystals are thin plates (5–10 µm thickness) can mitigate this phenomenon. Other structures solved at similar time points showed coherent alterations (data not shown), giving us confidence in the reproducibility of the results.

The two first time points structures (Fdh_1min and Fdh_1.5min) revealed for the first time three formate binding sites within the catalytic pocket of a Fdh. These binding sites can be relevant in terms of formate ingress towards the active site. *Dv*FdhAB variants will be produced to investigate the formate entrance pathway.

Unexpectedly, the first noticeable change upon formate incubation was the C5′ MGD2 swing. This movement seems to be independent of the metal site reduction changes. On the other hand, the distortion of the MGD2 dithiolene group seems to be progressive during the reduction process. These observations may explain the relevant role of MGD2 as a redox tuner of the metal center during turnover.

Globally, the reduction of the enzyme is propagating from the W active site to the essential catalytic residues (H193 and to some extent R441), then to the nearby residues (I191-P198 loop) and finally to the E443 and Q890 that interacts with MGD2 co-factor. Notably, full reversion to the oxidized state could be achieved after enzyme reoxidation by exposure to atmospheric oxygen, whereby the enzyme recovered its oxidized form without any apparent damage. The absence of damage upon reoxidation is probably crucial for protein stability and can be the reason why *Dv*FdhAB can be purified aerobically.

To conclude, we present here a time-resolved view featuring five different intermediate forms of the reduction reaction of *Dv*FdhAB with formate, along with the structural characterization of the oxygen-reoxidized enzyme. This provides further insights into how *Dv*FdhAB catalyzes the reversible oxidation of formate to CO_2_, but further experiments will be necessary to improve the time resolution and obtain more details about the catalytic mechanism. Tests are already being conducted in our lab with microcrystals being produced to perform time-resolved serial synchrotron crystallography experiments [10,11].

## 4. Materials and Methods

### 4.1. Expression and Purification of D. vulgaris FdhAB

*Dv*FdhAB was expressed and affinity-purified from *Desulfovibrio vulgaris* Hildenborough, as previously described [2]. The *Dv*FdhAB containing a Strep-tag was produced by homologous recombination and purified by affinity chromatography using a Strep-Tactin^®^ Sepharose^®^ resin (IBA Lifesciences). After protein purification, the sample buffer was exchanged and the samples were stored in 20 mM Tris-HCl buffer with 10% glycerol and 10 mM NaNO_3_, pH = 7.6. Routinely, protein concentration was determined based on U*V/V*is ($\varepsilon_{410\;\mathrm{nm}}=43.45\;{\mathrm{mM}}^{-1}{\mathrm{cm}}^{-1}$) and the purity of samples was judged by 12% SDS-polyacrylamide gel [2]. The samples were stored at −80 °C until further use.

### 4.2. Crystallization, Data Collection, Structure Solution, and Refinement

Crystallization of *Dv*FdhAB was performed under anaerobic conditions [2]. All the anaerobic experiments were performed in an anaerobic chamber under an argon atmosphere at <0.1 ppm of oxygen, and all the solutions were previously degassed and stored in the anaerobic chamber. All crystals were obtained using the hanging-drop vapor diffusion method, drops of 2 µL (1:1, protein:precipitant ratio) were set in 24 well plates (24 well XRL plate Molecular Dimensions) at 20 °C. *Dv*FdhAB at 10 mg/mL was crystallized in conditions with 22 to 26% PEG 3350 (*w/v*), 0.1 M Tris-HCl pH 8.0 and 1 M LiCl, with the addition of 0.2 µL of a dilution 1:500 from a stock of microseeds of *Dv*FdhAB to the drop and crystals appeared within 2 days. These crystals were then soaked for different incubation times (1 min, 1:30 min, 3 min, 5 min and 22 min), with 10 mM of sodium formate. To assess the reversibility of the redox state, protein crystals, co-crystallized with sodium formate were exposed to atmospheric oxygen for 12 min. After the soaking with sodium formate or with atmospheric oxygen all crystals were transferred into a cryoprotectant solution consisting of the precipitant solution supplemented with 20% (*v*/*v*) glycerol, and then flash cooled in liquid nitrogen.

X-ray diffraction experiments were performed on ESRF synchrotron (ID23-1 and ID30B beamlines) [12,13] and the data were processed with the programs autoPROC [14] and autoPROC featuring Staraniso protocol [15]. The structures were solved by molecular replacement with Phaser [16] from the CCP4 suite [17], using as search model the previously published as-isolated structure (PDB ID: 6SDR) for the reduction time-resolved experiment and the formate reduced structure (PDB ID: 6SDV) for the reoxidation experiment. The models were refined with iterative cycles of manual model building with Coot [18] and refinement with REFMAC5 [19]. The models were rebuilt with PDBredo [20] and the images produced with PyMOL [21]. Data processing and refinement statistics are presented in Appendix A.

### 4.3. Solution Kinetic Assays under Crystallization Conditions

Purified *Dv*FdhAB was used to perform kinetic assays under standard (as described in [2]) and crystallization conditions. For all conditions, turnover numbers were obtained under anaerobic conditions by following the absorbance of benzyl viologen (BV) at 555 nm (ε_555 nm_ = 12 mM^−1^ cm^−1^) using a U*V*/*V*is spectrophotometer (UV-1800, Shimadzu). For standard conditions, the assay was started after adding 20 mM formate to a cuvette containing 50 mM KPi Buffer pH 7.6, 2 mM BV, 1 mM DTT, and 1.4 nM pre-activated *Dv*FdhAB (pre-activation: 50 mM DTT, 5 min). The assay under crystallization conditions followed the same methodology, but with different buffer composition. Two solutions (A and B) were prepared and mixed in a 1:1 ratio to mimic the crystallization droplet environment. Solution A was 20 mM Tris-HCl pH 7.6, 10% glycerol, 10 mM NaNO3, and 1.4 nM *Dv*FdhAB without pre-activation. Solution B was 24% PEG3350, 100 mM Tris-HCl pH 8.0, and 1 M LiCl. To avoid protein precipitation, solutions A and B were mixed by adding solution B to solution A.

## Figures and Tables

**Figure 1 ijms-24-00476-f001:**
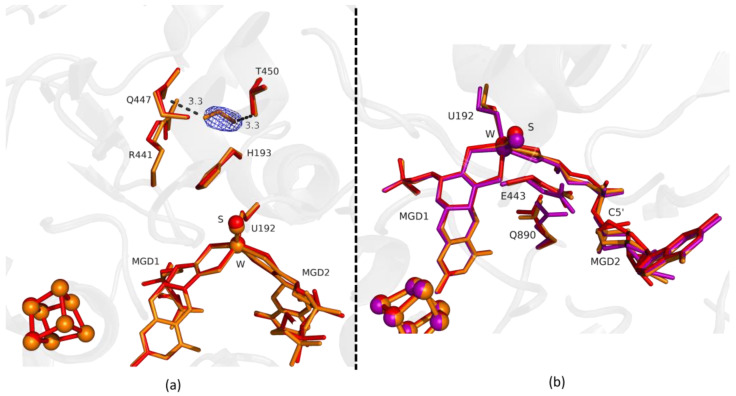
Superposition of *Desulfovibrio vulgaris* Formate dehydrogenase 1 (*Dv*FdhAB) as-isolated (red), formate-soaked Fdh_1min structure (orange), and formate reduced (violet) (in **b**) in two different orientations. (**a**) U192, H193, R441, Q447, T450 and the two Molybdopterin Guanine Dinucleotide (MGD) co-factors are shown as sticks, along with a formate molecule present in the formate tunnel of Fdh_1min. The 2fo-fc map is shown as a blue mesh, at 1.0 σ. The formate molecule is held by hydrogen bonds to side chains T450 Oγ and Q447 Oε (distance indicated in Å). (**b**) U192, E443, Q890 and the two MGD co-factors (note the C5′ atom of MGD2) are shown as sticks.

**Figure 2 ijms-24-00476-f002:**
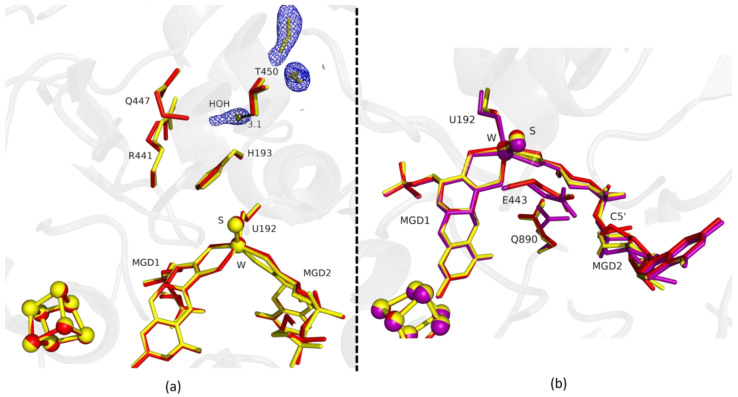
Superposition of *Dv*FdhAB as-isolated (red), formate-soaked Fdh_1.5min structure (yellow), and formate reduced (violet) (in **b**) in two different orientations. (**a**) U192, H193, R441, Q447, T450 and the two MGD co-factors are shown as sticks, along with two formate molecules and one water molecule present in the formate tunnel of Fdh_1.5min. The 2fo-fc map is shown as a blue mesh, at 1.0 σ. The distance between the water molecule and T450 Oγ is indicated in Å. (**b**) U192, E443, Q890 and the two MGD co-factors (note the C5′ atom of MGD2) are shown as sticks.

**Figure 3 ijms-24-00476-f003:**
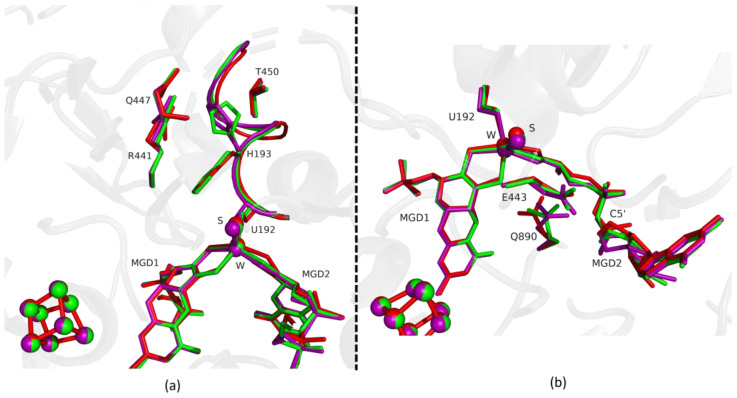
Superposition of *Dv*FdhAB as-isolated (red), formate reduced (violet) and formate-soaked Fdh_3min structure (green) in two different orientations. (**a**) U192, H193, R441, Q447, T450 and the two MGD co-factors are shown as sticks. Ribbon representation of loop I191-P198. (**b**) U192, E443, Q890 and the two MGD co-factors (note the C5′ atom of MGD2) are shown as sticks.

**Figure 4 ijms-24-00476-f004:**
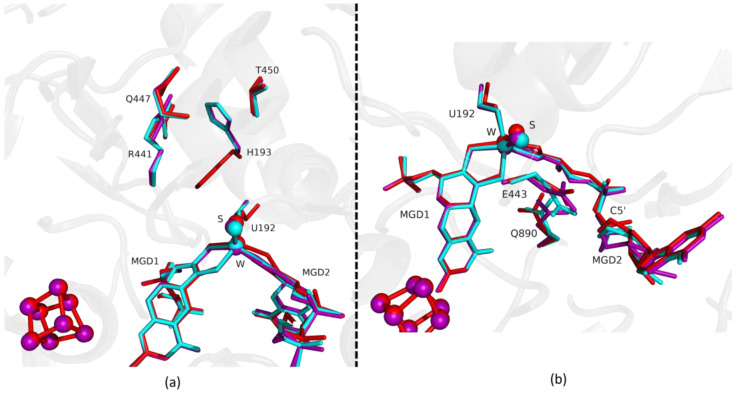
Superposition of *Dv*FdhAB as-isolated (red), formate reduced (violet) and formate-soaked Fdh_5min structure (light blue) in two different orientations. (**a**) U192, H193, R441, Q447, T450 and the two MGD co-factors are shown as sticks. (**b**) U192, E443, Q890 and the two MGD co-factors (note the C5′ atom of MGD2) are shown as sticks.

**Figure 5 ijms-24-00476-f005:**
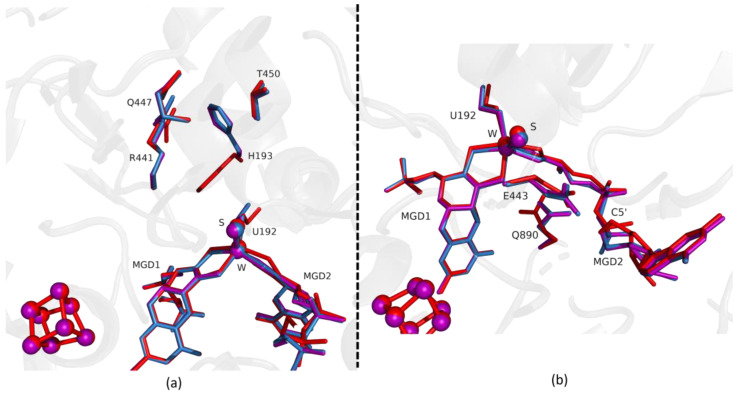
Superposition of *Dv*FdhAB as-isolated (red), formate reduced (violet) and formate-soaked Fdh_22min structure (dark blue) in two different orientations. (**a**) U192, H193, R441, Q447, T450 and the two MGD co-factors are shown as sticks. (**b**) U192, E443, Q890 and the two MGD co-factors (note the C5′ atom of MGD2) are shown as sticks.

**Figure 6 ijms-24-00476-f006:**
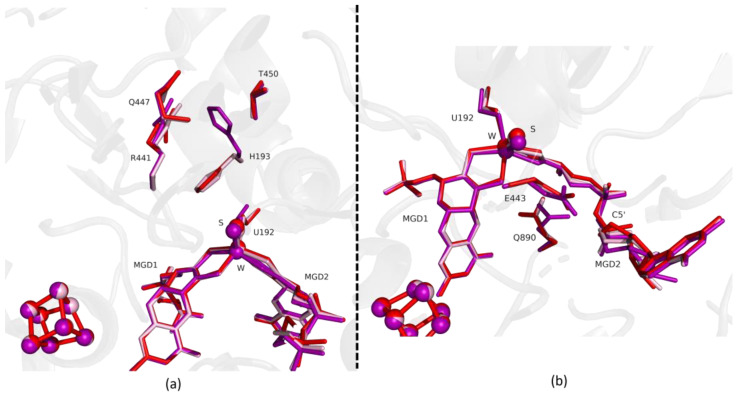
Superposition of *Dv*FdhAB as-isolated (red), formate reduced (violet) and reoxidized structure (light pink) in two different orientations. (**a**) U192, H193, R441, Q447, T450 and the two MGD co-factors are shown as sticks. (**b**) U192, E443, Q890 and the two MGD co-factors (note the C5′ atom of MGD2) are shown as sticks.

**Table 1 ijms-24-00476-t001:** Solution kinetic assays. *Dv*FdhAB kinetics under standard and crystallization conditions.

Condition	Pre-Activation	Turnover (s^−1^)	Relative Activity (%)
Standard	Yes	911 ± 113	100
Crystallization buffer	No	7 ± 1	0.8

## Data Availability

Protein structure coordinates and diffraction data for the structures present in this work are available at Protein Data Bank, under the accession codes: 8BQG, 8BQH, 8BQI, 8BQJ, 8BQK and 8BQL.

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
