# Peer review of "Tracking W-Formate Dehydrogenase Structural Changes During Catalysis and Enzyme Reoxidation"

_ijms, 2022, doi:10.3390/ijms24010476_

Round 1
Reviewer 1 Report
Vilela-Alver et al describe the structure determination of SeCys/W Formate dehydrogenases (Fdh) from Desulfovibrio vulgaris Hildenborough. Using soakings of different ligands under aerobic and anaerobic conditions, the author obtained snapshots of the protein in different states. Using the structural information obtained, the authors determine de conformational changes related with the reduction and oxidation of relevant molecular moieties during the reaction mechanism. The work in the paper is scientifically sound, and I therefore recommend publication with minor revisions.
Please find my suggestions for improvement below.
Is time-resolved the right term? The structures are more snapshots of different states that have been obtained at different soaking times. The structure do not provide insight into the reaction mechanism as such, but rather on the oxidation states of the cofactors and relevant amino acids.
The 1min and 1.5min time points show the appearance and subsequent disappearance of a formate molecule (lines 89-91 and 100-103)
- Is there an explanation for the disappearance of the formate molecule?
- It is mentioned that it was attempted to model the density with a water molecule, but unsuccesfully. Please provide more details on why this refinement failed
In the intermediate structure, there is a gradual shift between one and the next position, instead of two distinct conformations (lines 141-145)
- Please quantify the shift
- The density should show if two conformations exist or whether there is an intermediate position. Have you checked this? Can the sidechains be refined using multiple occupancies?
- It would be helpful to show the local density in a Supplmentary Figure
lines 149-150: The MGD2 is now superimposable with the reduced form (Figure 5b).
- Please quantify the superpositions with residue specific RMSDs, for example of intermediates forms with the fully oxidized and fully reduced forms. I would like to see this throughout the paper, eg lines 168-169, for relevant residues shown in figs 1-6. A table could be added with the calculated RMSDs
lines 245-246: staraniso is not a separate program, but rather a protocol used by AutoPROC
Reviewer 2 Report
This is an interesting continuation of a long term study on this curious metal-enzyme though I don't think it would be interesting for controlling the atmospheric CO2 as emphasised in the very first sentence of the introduction (lines 27-31)..
This is a good job, with great care in the analysis. Just a few questions and thinkings to do to deliver a better looking paper.
1-In all the figures, it should be nice to use the same complementary red/green/cyan colors than those adopted like Violet or yellow/violet (difficult to see the three chains altogether .. and I'm not daltonien) . Give also the characteristics of the difference F map leading to the density for the formate ion.
2- Paragraph 2.2 (line 158). I I guess that the crystals used for the reverse reaction don't come from the same well as those used in the previous kinetic. Would it be judicious to use some of the crystals named 22 mn (or more), as they are representing the end of the reaction, to really characterise a reversible process?
Line 243 :Do you include the soaking time in the reduction kinetics ?
- The enzyme has a selenocysteine (resid.192). I guess it comes from the expression system and not from a natural source. You may comment about any risk that this selenocysteine may influence the rate of the kinetics vs a classic cysteine?
Finally, before thinking about serial crystallography, not so evident to tackle, may I suggest to test CO2 (or O2) pressure crystallography as available at ESRF as to follow the tunnelling process within the enzyme?
Now you have coordinates of the reduction sets (0-1-1.5--3-5-22min), noway to present a morphing view (~video) of the reaction? That's would be great.. and informative about internal movements.
